# Impact of Stressful Events on Motivations, Self-Efficacy, and Development of Post-Traumatic Symptoms among Youth Volunteers in Emergency Medical Services

**DOI:** 10.3390/ijerph16091613

**Published:** 2019-05-08

**Authors:** Eleni Roditi, Moran Bodas, Eli Jaffe, Haim Y. Knobler, Bruria Adini

**Affiliations:** 1The Department of Emergency Management & Disaster Medicine, School of Public Health, Sackler Faculty of Medicine, Tel-Aviv University, Tel Aviv 6997801, Israel; eleniroditi8@gmail.com (E.R.); moranb@gertner.health.gov.il (M.B.); 2Israel National Center for Trauma & Emergency Medicine Research, The Gertner Institute for Epidemiology and Health Policy Research, Sheba Medical Center, Tel Hashomer, Ramat-Gan 5266202, Israel; 3Magen David Adom (the Israeli Red Cross), Tel Aviv 6706210, Israel; Eliy@mda.org.il (E.J.); Haim.knobler@gmail.com (H.Y.K.); 4Medical School, Ben Gurion University in the Negev, Beersheba 8499000, Israel; 5Hadassah-Hebrew University Medical School, Jerusalem, 9112102, Israel and Peres Academic Center, Rehovot 7610202, Israel

**Keywords:** EMS, youth volunteers, volunteering motivations, self-efficacy, post-traumatic symptoms

## Abstract

During the last decades, Israeli emergency medical services (EMS) personnel has been exposed to different potentially traumatic events, including mass terror attacks. The aims of the present study were to identify how potentially traumatic events affect young volunteers in their motivation to volunteer and their perceived self-efficacy while being at risk of developing post-traumatic symptoms. The final sample included 236 Magen David Adom (MDA, the "Israeli Red Cross") youth volunteers. The study evaluated their motivational factors for volunteering, perceived self-efficacy, participation in potentially traumatic events, and post-traumatic symptoms. Over two-thirds of the volunteers participated in a traumatic event on duty. Volunteers who were involved in potentially stressful events scored higher levels of post-traumatic symptoms, though still very low and subclinical. Nonetheless, participating in stressful events contributed to an increased sense of self-efficacy. No difference in post-traumatic symptom levels was observed between volunteers who partook in mass casualty incidents and those who did not. The results demonstrate that MDA youth volunteers may mostly benefit from participating in situations requiring the administration of emergency medicine, even stressful ones. They may help to find ways to empower the volunteers and increase their resilience.

## 1. Introduction

Provision of appropriate medical care to patients and casualties in routine as well as in mass casualty incidents (MCIs) depends on an effective system of emergency medical services (EMS). EMS can be defined as a system which provides emergency medical care by administrating the needed personnel, facilities and equipment for the best health services to the patients [1,2]. EMS involves emergency vehicles, such as ambulances and helicopters, as well as hired personnel and volunteers. One of the most important objectives of EMS is the evacuation of patients to the appropriate medical facility, according to their medical condition [2].

Non-profit organizations require human and financial resources for their long-lasting survival and thus, rely on young volunteers to supplement their needs [3]. Volunteering has been found to be beneficial, most specifically concerning the volunteers’ life- satisfaction, self-esteem, self-rated health, educational and occupational achievements, functional ability and decreased mortality [4].

EMS personnel, including volunteers, are regularly exposed to human pain and suffering, resulting from the situations they are exposed to while treating patients and casualties [5]. Traumatic events and chronic stress affect the EMS staff psychologically and may lead to the creation of varied reactions including loneliness, low self-efficacy, and even post-traumatic stress disorder (PTSD) [6,7,8]. Loneliness is the subjective social seclusion one experiences [9]. Self-efficacy refers to the level of self-reported perceptions regarding competence to achieve one’s ambitions [10]. PTSD is a psychiatric disorder that may appear among individuals that experience a traumatic event [11,12]. Symptoms of PTSD may include intrusive thoughts like re-experiencing the traumatic events, avoidance of places and activities that may remind of the trauma or hyper-arousal, such as sleep deficiency and being overly alert [13].

Thormar et al. [14] reported that volunteers have higher levels of mental health disorders compared to professional EMS workers. The negative psychological impact results from the exposure to varied stressful events, such as treatment of severe casualties, provision of cardiopulmonary resuscitation (CPR) procedures, or exposure to bodies of people who suffered violent death [15]. Youth EMS volunteers may be even more susceptible to psychological disorders [16]. Previous research among MDA youth volunteers found that terrorist attacks may cause a significant psychological burden on EMS youth volunteers, with conflicting results: Since the adolescents’ motives to volunteer vary, volunteers with certain motives retain their enthusiasm for volunteering despite the danger of developing PTSD symptoms. It was also found that some of the motives to volunteer correlated with a reduction in the level of post-traumatic symptoms [12].

Therefore, in contrast to the detrimental impact, trauma may also induce a positive impact on EMS personnel, for example: Improve the relationships of the volunteers or enhance their levels of self-efficacy, leading to post-traumatic growth (PTG) [17]. Social support was found to facilitate effective coping mechanisms [15]. Adolescents have been found to have PTSD symptoms long after a catastrophic earthquake (three years) while self-esteem was found to be a protective factor against PTSD symptoms among them [18].

Magen David Adom (MDA, the “Israeli Red Cross”) is Israel’s national emergency medical, disaster, ambulance, and blood bank service. MDA was founded by volunteers, and volunteering is an integral part of MDA. Approximately 4000 of MDA’s 14,000 volunteers are aged 15–18 years old and are described as youth volunteers [19]. MDA volunteers take part in numerous activities of the EMS, ranging from response to routine emergency calls, such as motor vehicle accidents (MVAs) or patients suffering from heart attacks, up to response to MCIs. Specifically, given the nature and level of risk posed by terror groups in Israel, MDA volunteers can be considered to be at a higher risk of exposure to terror-related MCIs [12]. To ensure continuity and well-being of the youth volunteers, it is important to understand what motivates them and how the varied events that they experience affect them, more specifically their inclinations to develop psychopathology (e.g., PTSD), feelings of loneliness, and self-efficacy levels.

The aim of this paper was to identify how different, potentially stressful events affect the motivations of adolescent EMS volunteers, their perceived self-efficacy, and their risk of developing post-traumatic symptoms.

## 2. Materials and Methods

### 2.1. Ethical Considerations

The study was approved by the Institutional Research Committee of MDA (approval number 118, from July 9, 2016).

### 2.2. Population and Sample

The study was based on data collected from 237 youth volunteers out of 280 volunteers (85% response) who participated in two courses conducted by MDA during the summer of 2016: Counselors’ training (23%) and mass casualty response training (77%). At the time of training, all volunteers were youth aged 16 to 21 years. The mean age was 16.46 (± 0.83 SD), with the majority of volunteers (52%, *n* = 123) aged 16. See Table 1 for a demographic overview of the sample.

### 2.3. Tools and Variables

To conduct the study, a questionnaire was designed, consisting of six main components: (A) Demographics (including age, country of origin, place of residence, level of religiosity (religious/ traditional/ secular), and seniority in MDA), (B) motivational factors for volunteering—31 factors adopted from Volunteer Functions Inventory [20] that were classified into three categories relevant to EMS volunteers: Humanitarian values (14 items, Cronbach alpha = 0.694), aspiration to become a medical professional (nine items, Cronbach alpha = 0.716), and fascination with the paramedic profession (eight items, Cronbach alpha = 0.736), (C) levels of post-traumatic symptoms measured by scales of hyper-arousal (eight items, Cronbach alpha = 0.737), avoidance (eight items, Cronbach alpha = 0.784), and intrusion (eight items, Cronbach alpha = 0.864), (D) perceived self-efficacy (six items, Cronbach Alpha = 0.885), (E) participation in potentially stressful events (six items, each measured individually and as a scale summating the amount of stressful events counted per participant), and (F) levels of loneliness (nine items, Cronbach alpha = 0.813). The specific scales used are listed below. All scales that were used in the study were previously validated and pilot-tested for applicability among youth MDA populations in former research that was conducted and published [12,16,21].

The motivational factors were measured by a questionnaire developed for the characteristics of youth volunteers in EMS frameworks. The factors were ranked by the respondents, using a Likert scale ranging from one to seven, with one indicating strongly disagrees and seven indicating strongly agrees.

Post-traumatic symptom levels and impact of traumatic events on the volunteers were measured using the impact of events scale (IES) developed by Horowitz et al. [22] that was previously modified to the characteristics of youth volunteers in EMS frameworks [12]. The IES scale is a self-measurement tool, designed to assess current subjective distress due to any particular traumatic event. It is one of the most internationally used scales for measurement of post-traumatic psychological effects [23,24]. The original IES scale measures the frequency of self-reported post-traumatic symptoms of intrusion and avoidance. The questionnaire that was used included the 15 original items of the IES scale and one additional question (a total of 16 items). In order to increase measurement sensitivity, answers were collected on a seven-point Likert scale ranging from one (strongly disagree) to seven (strongly agree), which were later re-categorized into the original IES categories: Answers 1 and 2 were grouped and recoded as zero ("not at all"), answers 3 and 4 recoded as one ("rarely"), answers 5 and 6 recoded as three (sometimes), and answer 7 recoded as five ("often"). Post traumatic level score was computed as the sum of answers provided by a respondent to all 15 items, therefore ranged from zero to 80. Levels of post-traumatic stress disorder (PTSD) symptoms were defined as follows: 44 and over: severe range, 26–43: moderate range, 9–25: mild range, 8 or lower: subclinical range [13,22]. The questionnaire also included six yes/no questions to evaluate the hyper-arousal based on the revised IES scale (IES-R) [13].

The level of loneliness and social isolation of the youth volunteers was measured through a revised version of the De Jong Gierveld loneliness scale [9], customized to the needs of the MDA volunteers and validated for use in Hebrew [25]. The scale consists of nine positively and negatively formulated sentences. The scores were first calculated individually, and then summed to calculate the overall score of loneliness. Upon reversing the scores for the negative statement into positive, the loneliness scale could range from 9 to 27. The volunteers ranked the answers depending on their applicability, ranging from one (not applicable), two (sometimes), to three (applicable). Levels of loneliness were defined as follows: Not lonely (24–27), moderately lonely (19–23), lonely (14–18), and extremely lonely (13 or less).

Self-efficacy was measured by six items, based on a Likert scale of one to six, one indicating “never” and six indicating “always”. The self-efficacy was computed based on the mean score of the six items. The questions encompassed the perceived problem-solving abilities of the youth volunteers, their successful ways of thinking, and the ability to achieve their goals.

Participation in stressful events was defined through six yes/no questions that allowed to identify the exposure to varied potentially traumatic situations, ranging from treating severely injured casualties in motor vehicle accidents (MVAs), provision of life-saving procedures, performing CPR, involvement in response to MCIs, and being exposed to dead patients. The participation in stressful events scores was computed as the summation of "yes" answers. Participants were also prompted to provide open text explanation if they wanted.

### 2.4. Statistical Analysis

The data was analyzed using SPSS 25. The normality of distribution was evaluated using Shapiro–Wilk test which showed that the population deviated from a normal distribution. Subsequently, all analysis was performed using non-parametric tests. Cronbach Alpha was used to test the reliability of scales. Spearman correlations were used to examine correlations between continuous variables. Friedman and Mann Whitney tests were used to test differences/variability among different target groups. Significance was defined at *p* < 0.05.

## 3. Results

### 3.1. Exposure to Traumatic Events and Dispositional Factors

Over two-thirds of the volunteers (*n* = 165; 69.9%) participated in a traumatic event while on duty: 110 (46.6%) participated in provision of CPR (non-MCI related), 125 (53.0%) were involved in providing medical care to severely wounded patients from motor vehicle accidents and a similar number of volunteers reported that they had seen or were involved in an event in which one or more patients died. Only 18 respondents (7.4%) reported partaking in an MCI. The mean score for stressful events participation was 2.58 (±1.53), with a majority (67.4%) reporting exposure to three or less stressful events (out of six).

The mean score for loneliness was 22.58 (± 4.34 SD). Almost half of the volunteers (*n* = 114; 48.3%) belonged to the "not lonely" category. Others were moderately lonely (*n* = 49;20.8%), lonely (*n* = 32; 13.6%), and extremely lonely (*n* = 8; 3.4%). The remaining 33 respondents (14.0%) had missing data for this scale.

### 3.2. Motivational Factors for Volunteering

The mean scores for the varied motivational factors for volunteering were all found to be high (more than five on a scale of seven). The mean scores were 5.09 (±0.96 SD) for fascination with the paramedic profession, 5.12 (±0.75 SD) for humanitarian reasons, and 5.38 (±0.87 SD) for aspiration to become a medical professional. Volunteers who participated in treating severely injured casualties from MVAs scored significantly higher than those who did not have the opportunity to provide medical care to such casualties on both the fascination with the paramedic profession (5.24 ± 0.89 versus 4.84 ± 1.05) and aspiration to become a medical professional (5.49 ± 0.89 versus 5.23 ± 0.87), according to Mann–Whitney test (*U* = 6836.00, *Z* = 2.855, *p* = 0.004 and *U* = 6476.00, *Z* = 2.048, *p* = 0.041, accordingly). Humanitarian values did not differ between these groups (*p* > 0.05).

Participating in treating severe casualties from MVA also contributed to a higher sense of self-efficacy (4.81 ± 0.84) compared to volunteers who did not have that experience (4.56 ± 0.82), according to Mann-Whitney U test (*U* = 5833.00, *Z* = 2.194, *p* = 0.028). Similarly, having experience in performing CPR also boosted volunteers’ perception of self-efficacy (4.81 ± 0.78) compared to those without such experience (4.56 ± 0.87), according to the same test (*U* = 5329.00, *Z* = 2.098, *p* = 0.036).

No significant differences in mean scores of motivational factors were found between volunteers who had or had not been exposed to MCIs, witnessed dead patients, performed CPR procedures, or participated in other potentially traumatic events (data not shown).

### 3.3. Levels and Correlates of Post-Traumatic Symptoms

A subclinical mean total score of post-traumatic symptoms (7.44 ± 10.51 SD) was found among the 208 volunteers who completed the questions concerning this component. Approximately 35% of the volunteers scored zero on this scale. The median score was 4.0. The mean score for intrusion was 2.78 (±5.67 SD), and for avoidance the mean score was 4.70 (±6.18 SD). Of the valid 214 answers, almost half of the volunteers (46.7%) reported at least one component of hyper-arousal (Table 2).

Volunteers that were involved in potentially stressful events scored higher levels of symptoms (i = 0.299, *p* < 0.001), albeit relatively rare. Having experience in performing CPR, providing medical care to seriously injured victims of an MVA, and witnessing death were also associated with elevated levels of symptoms (Table 3). Similar findings were found for all three sub-components of PTSD (intrusion, avoidance, and hyper-arousal), see Figure 1. However, having experience with CPR also increased volunteers’ perception of self-efficacy. Volunteers with CPR experience scored a self-efficacy mean of 4.81 (±0.78) as opposed to those without CPR experience who scored 4.56 (±0.87). This difference is statistically significant according to Mann–Whitney U-test (*U* = 5329.00, *Z* = 2.098, *p* = 0.036).

Volunteers who participated in treating severe casualties of MVAs, had significantly higher levels of post-traumatic symptoms compared to volunteers that were not required to provide medical care to such casualties. A similar trend of significance was found for the sub-components of the post-trauma scale (intrusion and avoidance), but no statistical difference was observed for the mean number of hyper-arousal effects reported (Figure 2). Experience with treating a severely wounded casualty of a MVA also contributed to the volunteers’ perception of self-efficacy, with those experienced reporting a mean score of 4.81 (±0.84SD) compared to those without such experience who scored a mean value of 4.56 (±0.82SD), according to Mann–Whitney U-test (*U* = 5833.00, *Z* = 2.194, *p* = 0.028).

Similar trends were observed for other stressful events, e.g., experiencing a traumatic event and witnessing the death of a patient. However, in contrast to the former stressor, the latter two were not associated with elevated levels of perceived self-efficacy (data not shown).

No statistical difference in PTSD symptom levels was observed between volunteers who partook in MCIs (8.12 ± 7.37 SD) and those who did not (7.29 ± 10.82 SD), according to Mann–Whitney *U*-test (*U* = 1.951.00, *Z* = 1.587, *p* = 0.112). Nevertheless, the data suggest that there is a statistical difference in the levels of avoidance between the groups with a mean score of 7.06 (±7.29 SD) for those who partook in a MCI and a mean score of 4.45 (±6.10 SD) for those who did not (*U* = 2044.00, *Z* = 2.105, *p* = 0.035).

### 3.4. Multi-Variant Analysis

Several correlations were observed between studied variables (Table 4). Of importance, the correlates of the post-traumatic symptoms’ mean score were the number of stressful events experienced by the volunteer and the number of hyper-arousal effects reported. Aspiration to become a medical professional and fascination with the paramedic profession were positively correlated with having stronger humanitarian values.

Based on the correlations reported, a linear regression analysis was performed in enter mode to predict PTSD symptoms’ mean score. Variables found to be associated with the dependent variable (i.e., hyper-arousal and the number of stressful events experienced), together with age, were introduced into the regression model. The model, which is statistically significant (F = 18.010, p < 0.001) and accounted for 22.7% of the total variance of the dependent variable, suggests that the only predictor of PTSD symptoms is hyper-arousal (*β* = 0.444, *p* < 0.001).

## 4. Discussion

This study describes how different, potentially stressful events may affect the motivations, perceived self-efficacy and the risk of developing PTSD among youth that volunteer in the Israeli EMS.

This study shows that participation in challenging situations, e.g., MCIs and trauma-bearing situations may enhance the motivation of youth to volunteer. Volunteers that actively participated in treating severe casualties and provided them with life-saving procedures were more fascinated with the paramedic profession and were more aspired to become medical professionals. This finding is in line with previous studies that found that the majority of first responders’ volunteers that experience stimulating events intend to retain their volunteering for a long period of time [26]. Ghaniyoun et al. [27] presented that anxiety is correlated with excitement and serves as psychological empowerment which enhances performance and organizational empathy.

Aligned with previous findings, the study identified that participation in stressful events may also have a detrimental influence, including a higher prevalence of PTSD symptoms and/or other mental health issues [6]. The youth volunteers in the study were found to have low levels of PTSD symptoms. EMS personnel in Karachi, Pakistan, were found to have a moderate level of PTSD [7], while in the Swedish EMS, 21.5% scored high levels [28]. Volunteers may be less trained and not as experienced in dealing with stressful situations, and thus, are more susceptible to adverse reactions. This assertion is supported by previous research reporting that volunteers that participated in treating casualties in varied stressful situations were afflicted with higher levels of PTSD compared to those who did not participate in such events [29,30].

Compton found that an unsuccessful CPR of loved ones administered by non-EMS workers or volunteers can be associated with PTSD symptoms [31]. Similar to the current findings, Genest et al. [32] identified that following CPR, ambulance volunteers suffer from psychological impacts, such as uncontrollable, involuntary thoughts, negative feelings and recurring mental images. Mathiesen et al. [33] found that CPR affected psychologically those citizens that provided it, and Kolehmainen et al. [11] found that internal medicine residents reported PTSD symptoms after participating in CPR. In line with those previous findings, MDA youth volunteers that provided CPR had higher mean scores of intrusion, hyper-arousal and avoidance compared to volunteers that had not provided such procedures. Such symptomatology raises the need to invest designated attention to the vulnerabilities of EMS volunteers to reduce the risk of their developing PTSD symptoms or other adverse conditions. Nevertheless, despite the drawbacks, experiencing stressful events, such as resuscitating a patient or treating a severely wounded MVA casualty also contributed to a higher perception of self-efficacy. This finding resonates with other studies performed in Israel and abroad that demonstrated the capability of hands-on experiences in promoting youth resilience and self-efficacy [34,35,36,37]. It is reasonable to assume that in the specific context of this study, being able to perform in stressful situations enhanced the youth volunteers’ confidence in their ability to efficiently cope with other unexpected or challenging events. It also reflected their competence to achieve one’s ambitions, ergo increasing their perceived self-efficacy.

MDA’s youth volunteers that were exposed to dead patients were found to have higher post-traumatic symptom levels, as compared to volunteers that were not exposed to bodies. These findings correlate with previous studies that found that exposure to dead victims significantly increased intrusion and avoidance symptoms [15,38].

In contrast to the findings of the current study in which no significant difference was detected in post-traumatic symptom levels of youth volunteers that had or had not participated in MCIs, Cetin et al. [39] found that volunteer rescue workers after natural disasters (earthquake) had significantly higher levels of intrusion, avoidance, and hyper-arousal mean scores. Similar findings were also reported by Thormar et al. [23] that found that disaster workers and community volunteers had higher levels of PSTD following a disaster. The lack of difference in mean scores of such symptoms among the Israeli youth volunteers that participated in MCIs may be related to the ongoing heightened level of alert that characterizes the Israeli society. Familiarity with a situation, even if a stressful one, may enhance resilience, that is the volunteers’ capacity to withstand and recover from hardships. For example, emergency medical teams in Portugal that were frequently exposed to traumatic car accidents had a low prevalence of PTSD [8].

The different motivational factors (humanitarian values, fascination with the paramedic profession, and aspiration to become medical professionals) were identified in the study to correlate with mean scores of perceived self-efficacy of the volunteers, albeit the first two were deemed non-significant following correction for multiple comparisons. This may be explained by a similar finding that was reported by Burger and Samuel [40], presenting the relationship between self-efficacy, psychological and social resources, and function in the varied systems. Combined with the findings concerning self-efficacy, the overall trend seems to suggest that partaking in a stressful situation contributes to attitudinal stances of youth volunteers by demonstrating their competencies to perform, and thus, empowering them. More so, it answers the need of many young individuals to be involved in what they perceive as “exciting and stimulating” activities, which further enhances their attraction with the medical professions. This may suggest that investing efforts in building the confidence of youth volunteers and strengthening their individual self-efficacy and image may deepen their commitment to the organization and sustain their volunteerism over time.

Contrary to other studies, mean scores of self-efficacy were found to be correlated with mean post-traumatic symptom levels, though the correlation was low and non-significant following multiple comparison corrections. Volunteers that participated in different potentially traumatic events were also found to have higher self-efficacy. Both Blackburn and Owen [41] and Nygaard et al. [42], found that higher self-efficacy was significantly related to lower PTSD levels among war veterans and survivors from natural disasters [37,38]. The surprising finding in the current study may be due to the continuous exposure of young volunteers to stressful events. Despite the belief of the youth in their own capacities, the ongoing involvement with challenging situations may tax their mental resources and raise the risk of developing PTSD symptoms.

### Limitations

This study has several limitations. First, the study was based on a self-reporting questionnaire that was given to volunteers participating in MDA courses, ergo the tool is susceptible to reporting biases. In addition, the tool is based on a retrospective recollection of attitudes, and therefore, is subjective to memory biases. Second, while the questionnaire utilized several validated tools, some of those were presented on scales different from the original. This was done to increase research sensitivity and allow for an increased variance of responses. Third, Israeli volunteer youth, especially in the context of emergency medical services, may present different characteristics than their peers around the world and/or the general population. Generalization of conclusions should be done with caution.

## 5. Conclusions

The present study aimed to identify how various potentially stressful events affect the motivations, perceived self-efficacy and risk for developing PTSD among volunteers. As presented, while the humanitarian values of Israeli EMS volunteers remain unchanged after experiencing stressful events, their fascination with the paramedic profession, their aspiration to become medical professionals and their perceived self-efficacy increase. Nonetheless, in tandem with the positive impact, the risk of developing post-traumatic symptoms also increases after experiencing such events.

The lessons learned from this study can contribute to EMS organizations, as well as other organizations looking to expand their teams of volunteers, in establishing and retaining such crucial personnel. For example, organizations could attract volunteers by investing in increasing their self-efficacy, but in parallel, they need to take measures to protect the volunteers from developing PTSD symptoms. To achieve this, the volunteers must receive support following exposure to stressful events, to decrease their risk of developing psychological symptoms and encourage their long-term volunteering. This is in addition to rigorous screening of volunteers prior to recruitment and a well-developed training program to maximize volunteers’ capacities to endure foreseen hardships. It is important to explore ways to empower the volunteers and increase their resilience, so that they be less susceptible to adverse reactions and more capacitated to positive function.

Considering the importance of recruiting volunteers to first responder bodies, it is recommended that further research be initiated to explore how other types of potentially stressful events impact on motivational factors, self-efficacy, and adverse reactions including PTSD among the volunteers. Such findings could also contribute towards an understanding of how perceived self-efficacy may positively affect the resilience to withstand stressful events.

## Figures and Tables

**Figure 1 ijerph-16-01613-f001:**
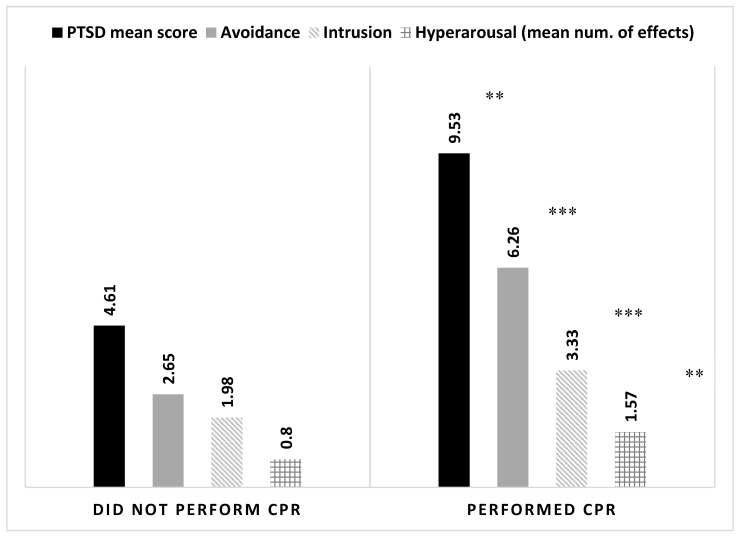
Mean levels of PTSD mean score and sub-components according to experience with provision of cardiopulmonary resuscitation (CPR) by MDA youth volunteers (*N* = 236). Statistical differences established with Mann–Whitney U-test. ** *p*-value < 0.01 *** *p*-value < 0.001.

**Figure 2 ijerph-16-01613-f002:**
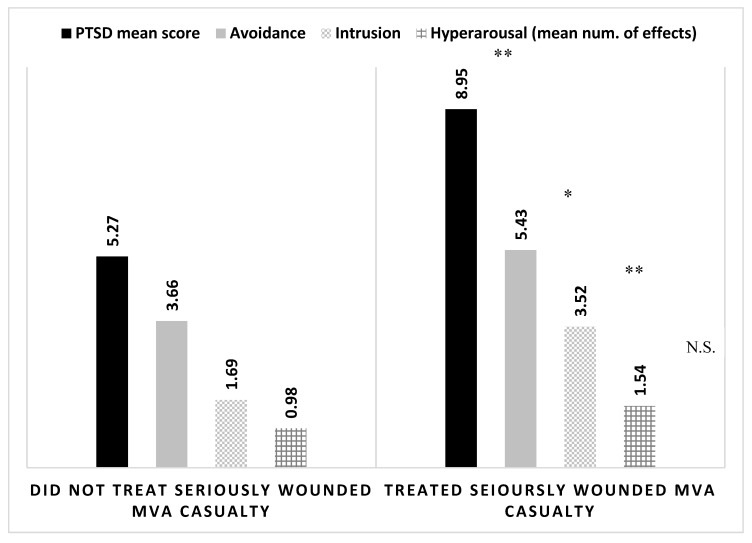
Mean levels of PTSD mean score and sub-components according to experience with treating a seriously wounded MVA casualty by MDA youth volunteers (*N* = 236). Statistical differences established with Mann–Whitney U-test. * *p*-value<0.05 ** *p*-value < 0.01 MVA = Motor Vehicle Accident.

**Table 1 ijerph-16-01613-t001:** Demographic characteristics of the study population.

Demographic Variable	Categories	Frequency (%)
Country of birth	Israel	221 (93.6%)
	Other (Argentina, France, Mexico, USA)	4 (1.6%)
	Missing	11 (4.7%)
Age	15	6 (2.5%)
	16	123 (52.1%)
	17	66 (28.0%)
	18	10 (4.2%)
	21	3 (1.3%)
	Missing	28 (11.9%)
Level of religiosity	Secular	115 (48.7%)
	Religious/Traditional	96 (40.7%)
	Missing	25 (10.6%)
Type of MDA Course participated	Counselors	54 (22.9%)
	Mass casualty response vehicle	182 (77.1%)
Seniority in MDA	1 year	162 (68.6%)
	2 years	57 (24.2%)
	Missing	17 (7.2%)

MDA = Magen David Adom (Israel’s EMS).

**Table 2 ijerph-16-01613-t002:** Frequency of hyper-arousal effects among MDA youth volunteers (*N* = 236).

Hyper-Arousal Effect	Valid Responses (n)	Frequency (%)
Sweating, accelerated breathing or pulse (w/o physical reason)	186	10 (5.4%)
Agitation	186	16 (8.6%)
Sleeping disorders	186	18 (9.7%)
Notable behavioral change (commented to you by others)	185	20 (10.8%)
Difficulties falling asleep	214	22 (10.3%)
Dreams about a stressful event	214	39 (18.2%)
Uncontrollable thoughts about a stressful event	186	50 (26.9%)
Recurring mental images from a stressful event	185	55 (29.7%)
Were the above-mentioned hyper-arousal effects associated with a specific traumatic event?	172	50 (29.1%)

**Table 3 ijerph-16-01613-t003:** Effect of experiencing stressful events on levels of PTSD among MDA youth volunteers (N = 236).

Stressful Experience (n)		PTSD Mean Score (±SD)	Mann-Whitney U	Standardized Z Score (*p*-Value)
Performing CPR	Yes	9.53 (±11.18)	6487.00	4.439 (<0.001)
	No	4.61 (±8.90)		
Providing medical care to seriously injured victims from a MVA	Yes	8.95 (±11.73)	6262.00	2.597 (0.009)
	No	5.27 (±8.11)		
Participating in a non-MCI traumatic event	Yes	8.20 (±10.22)	4826.00	3.396 (0.001)
	No	4.69 (±11.30)		
Witnessing death of a patient	Yes	8.86 (±10.73)	6112.00	3.346 (0.001)
	No	4.91 (±7.82)		

**Table 4 ijerph-16-01613-t004:** Spearman correlations of PTSD, motivational, and dispositional factors among MDA youth volunteers (N = 236^+^).

	2	3	4	5	6	7	8
1. PTSD mean score	0.299^***^	0.452^***^	–0.057	–0.023	0.102	0.003	0.170^*^
2. Number of stressful events		0.193^**^	0.007	−0.083	0.060	0.091	0.176^*^
3. Hyper-arousal (num. of effects)			−0.240^**^	−0.094	−0.015	−0.080	0.003
4. Loneliness				0.098	0.114	0.176^*^	0.297^***^
5. Humanitarian values					0.480^***^	0.393^***^	0.144^*^
6. Aspiration to become a medical professional						0.404^***^	0.202^**^
7. Fascination with the paramedic profession							0.185^*^
8. Self-efficacy							

^+^ Maximum missing per item: 33. ^*^ Correlation significant at 0.05 level (two-tailed), non-significant following multiple comparison correction (adjusted p-value = 0.007); ^**^ Correlation significant at 0.01 level (two-tailed); ^***^ Correlation significant at the 0.001 level (two-tailed).

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
