# Peer review of "Impact of Stressful Events on Motivations, Self-Efficacy, and Development of Post-Traumatic Symptoms among Youth Volunteers in Emergency Medical Services"

_ijerph, 2019, doi:10.3390/ijerph16091613_

Round 1
Reviewer 1 Report
This is a highly readable and informative paper. The authors made many critical and relevant points that should be carefully considered by programs recruiting volunteers to first responders. Really, a pleasure to read.
Author Response
We thank the reviewer for the comment.
Reviewer 2 Report
The paper identifies how potential traumatic events affect Israeli Red Cross youth volunteer attitudes and self-perceptions. The contribution of the paper is its use of empirical data in assessing a specific cohort of young volunteers who are exposed to traumatic events. The focus on young volunteers adds to the extant literature. The basic findings of the study are that exposure to trauma had a significant effect on volunteers in terms of their interest in the medical profession and aspiration to become a medical professional. An important additional finding is that exposure to trauma increased PTSD occurrence. Differences also are detected with regard to efficacy defined as self-reported competence to achieve ambitions, solve problems and think critically. Strengths of the paper are its ability to differentiate between stress exposed and non-stress exposed groups of youth volunteers. I believe the paper is methodologically sound, supports other studies, and adds to the literature in terms of the Israeli setting and age group. I think improvements can be made by:
1. Noting that volunteer are already a self-selected group and may not be representative of the general population
2. Further explicate the concept of self-efficacy focusing on the perception dimension of this concept and provide some rationale for why these scores may have increased.
3. Further speculation is called for in regard to the motivational factors. It should be recognized that volunteers may already be highly motivated; speculation of why exposure to trauma increases fascination levels and professional ambition would be helpful.
4. Reporting that trauma exposure PTSD scores are subclinical should be supplemented by a discussion of the clinical threshold and levels in the general population.
5. Clarify between motivation to volunteer (an event prior to experiencing trauma) and post-trauma perceptions. Survey seems to tap the latter dimension not the pre-volunteer perceptions of the general youth population.
6. The concept of resilience should be further defined.
7. Conclusions seem to focus on increased self-efficacy and post PTSD support. The paper could consider that efficacy is found as a benefit so there would be less need to increase efficacy, only to publicize it as a recruitment tool. The paper mentions post trauma support but neglect the role of training and screening of volunteers.
8. Authors might consider a new title since the constructs of motivations and self-efficacy are not as well developed as they can be.
Author Response
1. Noting that volunteer are already a self-selected group and may not be representative of the general population
We thank the reviewer for this comment. This is explicitly mentioned in the limitation clause. Nonetheless, we add that they may differ also from the general population.
2. Further explicate the concept of self-efficacy focusing on the perception dimension of this concept and provide some rationale for why these scores may have increased.
In the introduction it is stated that "Self-efficacy refers to the level of self-reported perceptions regarding competence to achieve one-self's ambitions". We added on Page 12 the following text to the relevant section of the discussion: " It is reasonable to assume that in the specific context of this study, being able to perform in stressful situations reflected volunteers' competence to achieve one-self's ambitions, ergo increasing their perceived self-efficacy."
3. Further speculation is called for in regard to the motivational factors. It should be recognized that volunteers may already be highly motivated; speculation of why exposure to trauma increases fascination levels and professional ambition would be helpful.
Two of three motivational factors were eventually non-significant in multiple comparison, so we cannot state for a fact that they are increased through participation in stressful events. Nonetheless, we highlighted the discussion on page 13 with the following text: "Combined with the findings concerning self-efficacy, the overall trend seems to suggest that partaking in stressful situation contributes to attitudinal stances of youth volunteers by demonstrating their competencies to perform and thus empowering them."
4. Reporting that trauma exposure PTSD scores are subclinical should be supplemented by a discussion of the clinical threshold and levels in the general population.
The categorization of the scores as "subclinical" is suggested by the validated tool we used ("Impact of Events Scale"). It proposes that scores below 8 should be regarded as subclinical in terms of PTSD psychopathology.
With regards to PTSD levels in the general population - according to Greene, Neria & Gross (2016), who systematically reviewed the literature for prevalence rates of PTSD in the population: "In Israel, none of the studies attempted to collect a representative primary care sample. Therefore these prevalence rates found in this review cannot be generalized to the primary care population as a whole". Therefore, suggesting the data for the population will prove difficult with the absence of reliable data.
Greene, T., Neria, Y., & Gross, R. (2016). Prevalence, detection and correlates of PTSD in the primary care setting: A systematic review. Journal of clinical psychology in medical settings, 23(2), 160-180.
5. Clarify between motivation to volunteer (an event prior to experiencing trauma) and post-trauma perceptions. Survey seems to tap the latter dimension not the pre-volunteer perceptions of the general youth population.
Motivation to volunteer was assessed based on three categories: humanitarian values, aspiration to become a medical professional, and fascination with the paramedic profession. In this sense, "motivation to volunteer" essentially answers the question "what do they volunteer for?". We wanted to assess the effects of participating in stressful event on the motivation (reasons) to volunteer.
6. The concept of resilience should be further defined.
We added on page 12 the text: "that is the volunteers capacity to withstand and recover from hardships" to explain the term "resilience"
7. Conclusions seem to focus on increased self-efficacy and post PTSD support. The paper could consider that efficacy is found as a benefit so there would be less need to increase efficacy, only to publicize it as a recruitment tool. The paper mentions post trauma support but neglect the role of training and screening of volunteers.
We added the following text to the conclusion section: "This is in addition to rigorous screening of volunteers prior to recruitment and a well-developed training program to maximize volunteers' capacities to endure foreseen hardships."
8. Authors might consider a new title since the constructs of motivations and self-efficacy are not as well developed as they can be.
We hope the changes made to the manuscript and the above explanations are better fitting with the title.

Reviewer 3 Report
I feel this is an excellent manuscript. It is highly relevant not only in Israel but for other countries as well. Your methods are very well explained as well as you statistics. Your tables are well done and easily understood standing alone. Your writing is very professional and clear.
My only concern is this. Although only a few respondents in your study showed any significant concerns, such as loneliness, was there any intervention for those individuals? Or was each respondent anonymous?
Author Response
We thank the reviewer for his/her comments. In response to their concern we would like to state the following: the study was conducted completely anonymously, in line with the ethical committee's approval. Therefore, no individual follow ups were possible in response to specific questionnaires. Nonetheless, MDA routinely operates a mechanism that allows supervisors to track volunteers who are in need of psycho-social support and provide necessary assistance when it is needed.
